# The Impact of Public, Private, and Out-of-Pocket Health Expenditures on Under-Five Mortality in Malaysia

**DOI:** 10.3390/healthcare10030589

**Published:** 2022-03-21

**Authors:** Renuka Devi Logarajan, Norashidah Mohamed Nor, Abdalla Sirag, Rusmawati Said, Saifuzzaman Ibrahim

**Affiliations:** School of Business and Economics, Universiti Putra Malaysia (UPM), Serdang 43400, Selangor, Malaysia; norashidah@upm.edu.my (N.M.N.); siraga87@gmail.com (A.S.); rusmawati@upm.edu.my (R.S.); saifuzzaman@upm.edu.my (S.I.)

**Keywords:** child, under-five, health expenditure, public, private, out-of-pocket

## Abstract

Health financing in Malaysia is intensely subsidised by public funding and is increasingly sourced by household out-of-pocket financing, yet the under-five mortality rate has been gradually increasing in the last decade. In this context, this study aims to investigate the relationship between public, private, and out-of-pocket health expenditures and the under-five mortality rate in Malaysia using the autoregressive distributed lag (ARDL) estimation technique, whereby critical test values are recalculated using the response surface method for a time-series data of 22 years. The findings reveal that out-of-pocket health expenditure deteriorates the under-five mortality rate in Malaysia, while public and private health expenditures are statistically insignificant. Therefore, an effective health financing safety net may be an option to ensure an imperative child health outcome.

## 1. Introduction

The World Health Organisation (WHO) [1] highlighted that USD 8.3 trillion was spent on health globally in 2018, representing close to 10% of the global Gross Domestic Product (GDP). Specifically, high-income countries spent 8.2% of GDP on health, followed by low-income countries and upper-middle income countries with 6.4% and 6.3% of GDP on health, respectively, and the lowest share was from lower-middle income countries with 4.8%. Globally, health spending was also increasingly reliant on public funding, which accounted for 59% of total spending on health in 2018, amounting to USD 4.9 trillion. The balance of 41%, or USD 3.4 trillion, is private health spending, of which a larger share was from household out-of-pocket health spending, compared to private health expenses from insurance and corporations. In fact, the sources of health spending also differ across the country’s income groups. Low-income and lower-middle income countries depend 40% on out-of-pocket health spending, as the public health spending was only 21% and 35%, respectively. While in upper-middle income and high-income countries, the largest source was from the public sector, which accounted for 38% and 48% separately. The second largest share was out-of-pocket health expenditure, at 35% of total health spending for upper-middle income countries, and private health expenditure, at 31% for high-income countries.

Similarly, in Malaysia, being an upper-middle income country, the government undertakes an active role in the country’s overall social and economic development, including the health sector, mostly through Federal Government funding. Even though total expenditure on health increased from RM8556 million in 1997 to RM60,147 million in 2018, it is still relatively small at 4% of GDP in 2018, compared to an average of 6% of GDP spent on health in upper-middle income countries.

As for Malaysia, health financing is intensely subsidised by public funding, which peaks at 51% (2018) of total health spending and is sourced by household out-of-pocket financing at 35% (2018). While the private health expenditure remains very much lower at 14% of total health spending, similar to lower-middle income countries, rather than being on par with peer countries within the upper-middle income country group, which averaged at 26%. A similar pattern was noted from 1997 to 2018, where public health financing remained higher than private funding. In fact, the share of public health expenditure and out-of-pocket health financing has been in a fluctuating trend for a decade since 1997. After the financial crisis in 2008–2009, the share of public health expenditure decreased, while out-of-pocket health spending steadily increased until 2018. During the same period (1997 to 2018), private health expenditure plateaued. Overall, per-capita health spending in Malaysia, which was RM1857 (approximately USD 450) in 2018, remains below the average for upper-middle income countries, which was USD 466 in 2018. The global health expenditure as well as in Malaysia are as in Figure 1 and Figure 2.

Another substantial aspect is that, currently, the world population consists of 7.6 billion people, and 30% of the world population, or 2.3 billion, are children. Similarly, in Malaysia, the proportion of children is also about 30% (9.2 million) of the total population. Within the child population, about 30% are aged below five (the statistic mentioned in this paragraph is based on the United Nations, Department of Economic and Social Affairs, Population Division (2019). World Population Prospects 2019, Online Edition. Rev. 1).

Globally, the under-five mortality rate (the probability of dying between birth and five years of age per 1000 live births) has decreased by 60%, from an estimated rate of 93 deaths per 1000 live births in 1990 to 38 deaths per 1000 live births in 2019, but it is still far from reaching the SDG targets of 25 deaths per 1000 live births by 2030. In 2019, an estimated 5.2 million under-five child deaths occurred, which is equivalent to about 14,000 child deaths each day before reaching their fifth birthday [2].

As for Malaysia, despite being an upper-middle income country, the under-five mortality rate has been reduced by 50% from 16.8 (1990) to 8.8 (2018) per 1000 live births, lower than the targets of the SDG target. Therefore, Malaysia’s rates are being compared to the corresponding rates in countries such as Singapore, the Republic of Korea, and Japan, where the under-five mortality rate is lower than Malaysia’s rate and those countries have recorded a decreasing trend [3]. Specifically, under-five deaths in Malaysia have been consistently high, with more than 4000 deaths on average since 2010 (source: Department of Statistics Malaysia). Malaysia’s under-five mortality rate, in comparison with the global rate, is shown in Figure 3.

In terms of the economy, even though Malaysia has also recorded sustainable real GDP growth with low inflation and full employment; however, according to the Malaysian Well-being Index, (developed by the Department of Statistics, Malaysia) the social well-being progressed at a slower pace compared to economic well-being.. The Index also revealed that health is one of the contributing factors to the fluctuation in a decreasing trend for social well-being since 2012. This indicates that Malaysia’s progress will not only be measured in terms of growth and wealth, but also in its ability to ensure the well-being of children in Malaysia. The wellbeing of children is vital, not only for sustainable development [4], but it also serves as a benchmark for performance comparison among countries, and it has been established that Malaysia is ranked lower in terms of children’s wellbeing compared to some economically advanced countries in Asia [5,6]. Therefore, although health financing in Malaysia is intensely subsidised by public funding and increasingly sourced by household out-of-pocket financing, yet Malaysia only devotes about 4% of GDP to health, which is still relatively low compared to upper-middle income countries. As such, failing to sufficiently recognise the impact of health expenditure will undermine efforts to increase interventions to improve child health outcomes.

Given this background, this study seeks to assess the effectiveness of the three types of health financing, namely, public, private, and out-of-pocket, as well as other macroeconomic factors, on improving the health of children under the age of five in Malaysia. Furthermore, the findings of this study may stimulate child-related policy formulation on health financing, in anticipation of improved healthcare. As such, this paper shall contribute to the existing analyses within the framework of health financing and health outcomes, particularly in Malaysia. Thus, the objective of this paper is to examine the impact of public, private, and out-of-pocket health expenditures on the under-five mortality rate in Malaysia. The remaining sections of the paper include the literature review, research methodology, results, and discussion, as well as the conclusion of the study.

## 2. Literature Review

The WHO [7] states that Universal Health Coverage is achieved when every individual has access to the health services as needed without any form of financial deprivation. As public spending on health is the main means of achieving universal health coverage, countries are undertaking various initiatives to enhance the financial resources for health. It was also reported that increasing public spending on health does not continuously create greater access to health services. The 2017 Global Monitoring Report on tracking universal health coverage emphasised that only about 50% of the global population has access to health services and about 800 million people spend 10% of their income on health care purposes, either for themselves or their family, especially children [7].

A large and growing body of literature has investigated the effect of public health expenditure on health outcomes, especially in developing countries where the prevalence of health-related challenges is still high. For instance, some recent studies have consistently revealed that public health expenditure significantly reduces the child mortality rate in Nigeria [8,9] and Ghana [10]. Specifically, in Nigeria [8], it was found that increases in the urban population decrease infant mortality rates, while per-capita income has no significance. Similarly, public health expenditure was also found to be effective in reducing the infant mortality rate among West African countries [11] and the under-five mortality rate in developing countries [12,13]. In addition, immunization, female literacy, improved water sources, and good sanitation are among other variables that hold a vital role in reducing child mortality rates. While, for the case of Malaysia, based on the autoregressive distributed lag model for time series data from 1984 to 2009, Riayati and Junaidah [14] reported that there was no cointegration among variables, namely, infant mortality and under-five mortality rate, public health expenditure, income level, corruption, and government stability.

Comparatively, a panel data analysis with a fixed effects model of BRICS nations—Brazil, Russia, India, China, and South Africa, by Kulkarni [15], differs by implying that higher public expenditure and out-of-pocket expenditure, as well as an increase in GDP per-capita, lead to a higher infant mortality rate. Similarly, in Sub-Saharan African countries, public health expenditure is not in favour of decreasing under-five and infant mortality, but urbanisation was found to be associated with a reduction in the mortality rates [16]. Considerably, Dhrifi [17] found a significant positive effect of public health expenditure on infant mortality for high-income countries, while for lower-, lower-middle, and upper-middle income countries, public health spending is not significant. As such, among the OECD countries, apart from public health spending, an increase in income per-capita also decreases the under-five mortality rate [18]. While Tejada [19] highlighted that allocating more public health spending reduces the negative impact of low GDP per-capita, high inflation, and unemployment rates on child mortality rates, particularly in low- and middle-income nations. Basically, unemployment, child poverty, material deprivation, and income inequality reveal a worst-off situation and impact negatively on child health [20,21].

Interestingly, based on the system generalised method of moments estimation technique, Ssozi and Amlani [22] revealed that even though all sources of health funding, which includes development assistance, government, private, and out-of-pocket, improve both infant and under-five mortality rates in 43 countries in Sub-Saharan Africa, but specifically public health expenditure, has the highest statistical significance in improving child health. Separately, three different studies [23,24,25] focused on total health expenditures, which comprises government and private health expenditures. Based on fixed effects estimation on panel data of 40 Sub-Saharan African countries, Ashiabi et al. [23] found that public health expenditure is inversely and significantly related to infant and under-five mortalities. Together, per-capita income, female literacy, and improved water sources are all significant in explaining child health, except for private health expenditure, which was found to be insignificant. Moreover, Bein et al. [24] revealed a negative relationship between total healthcare expenditures and the number of neonatal, infant, and under-five deaths for eight East African countries. While Boachie et al. [25] highlighted that public health expenditure and income were significant in improving child mortality rates in Ghana, when private health expenditure was included in the regression, the public expenditure was insignificant while the private contribution indicated a positive child health outcome.

Similarly, Raeesi et al. [26] investigated the effects of both private and public health expenditure on infant mortality rates and under-five mortality rates among 25 countries with different health care systems over the span of 15 years. It was proven that, apart from per-capita income, public health expenditure is more prominent in countries with national health care systems, while private health expenditure was found to be substantially important in countries with insurance health care systems. Alternatively, Dhrifi [27] explained that even though total health expenditure, which consists of both public and private funding, reduces the under-five mortality rate, it is detailed that public health spending has a strong impact in low and middle-income countries, while private health spending tends to be effective in higher-income countries. In addition, GDP growth, a reduction in poverty, and urbanisation also reduce the under-five mortality rate. Contrarily, based on the system generalizsed method of moments analysis of 195 countries, Ray and Linden [28] revealed that public health expenditures reduce the infant mortality rate in high-income countries, while private health expenditures are insignificant in both low- and high-income countries.

## 3. Methodology and Data

The theoretical foundation for this study is based on the health capital model by Grossman [29] as in Equation (1), whereby an individual’s stock of health, which is considered as an endogenous variable, depreciates with the age factor. Thus, investments in the form of improving health, able to retain or increase the stock of health.
(1)$$Ht=F\left(X_{t}\right)$$
where $H_{t}$ is the health outcome, while $X_{t}$ is all the inputs that determine the health outcome, which includes income, education, as well as health care inputs such as medical care.

Based on recent studies [8,9,11,12,25], the health capital model, as in Equation (1) is respecified by including the health expenditure ($HE_{t})$ and other control variables ($X_{t}$), as follows:(2)$$H_{t}=F\left(HE_{t}\;,\;X_{t}\right)$$
(3)$$H_{t}=\beta_{0}+\beta_{1}LHE_{t}+\beta_{2}X_{t}+{µ}_{t}$$

Therefore, in exploring the impact of health expenditures, namely, public, private, and out-of-pocket health expenditures, on the health outcome of children under the age of five, the following Equation (4) is derived:(4)$$LU5MR_{t}=\beta_{0}+\beta_{1}LHE_{t}+\beta_{2}LGDPC_{t}+\beta_{3}LUEMP_{t}+\beta_{4}LUPOP_{t}+{µ}_{t}$$
where, the child health outcome is measured by under-five mortality rate, U5MR. While, the health expenditure, HE is the main explanatory variable of child health outcome. Since the objective of this paper is to examine the impact of public, private, and out-of-pocket health expenditure on the under-five mortality rate in Malaysia. Thus, there will be three separate models, whereby the  HE will be represented by the public (PUH), private (PRH), and out-of-pocket (OOP) health expenditures separately, in order to assess the impact of each type of health expenditures on the under-five mortality rate. Other control variables are gross domestic product per-capita (GDPC), unemployment (UEMP), and urban population (UPOP). The L is the natural logarithm, µ represents the error term, and the subscript t denotes time.

Based on the standard classification of the WHO, the dependent variable, U5MR, is the probability of death by age 5 per 1000 live births, and it is widely recognised to measure the health status of children. The public health expenditures, PUH, refers to the Federal government’s funding, which consists of operating and development health expenditures such as for curative care, medical goods, hospital facility development, education, and training of health personnel as well as research and development in health. The private health expenditures, PRH, refers to private sector sources of financing mainly from insurance and corporations. The out-of-pocket health expenditures, OOP, refers to health related expenses borne by household or individual. In literature, health spending be it public [9,12], private [25], or out-of-pocket [22] is expected to improve child health. As for GDPC, it is used to measure the aggregate income level of a country and is an important control variable in determining the health status of a country. While UEMP refers to the total number of individuals in the working-age group between 15 and 64 years, who did not work but are interested in working or seeking a job. Whereas UPOP refers to the total population living in urban areas in Malaysia. The control variables, namely, GDPC, UEMP, and UPOP  are included in the model to avoid the problem of functional specification error in the model as well as to further contribute to knowledge by including relevant variables [12]. Generally, GDPC [11,23] and UPOP [8,16] are expected to improve child health, while UEMP [19,20,21] worsens.

Data for all the variables used in this study are obtained from publicly available data sources from Malaysian Government agencies, namely, the Department of Statistics, Ministry of Health and Economic Planning Unit, Prime Minister’s Department, as well as World Bank Open Data repository. The dataset covers the period between 1997 and 2018. Series in absolute values are fetched into EViews 11 statistical package and generated in the form of log. The EViews 11 software was used to run the relevant tests for all the variables in the model estimation. The descriptive statistics of the variables are presented in Table 1.

As for method, the autoregressive distributed lag (ARDL) bound test approach developed by Pesaran and Shin [30] as well as Pesaran et al. [31] is employed in this study, specifically for three main justifications. Firstly, the ARDL method is applicable regardless of whether the regressors are integrated of order 0, I(0), integrated of order one, I(1), or a combination of I(0) and I(1). Secondly, the ARDL method has better properties and generates reliable coefficients for a small sample size, approximately between 30 and 80 [32]. Thirdly, the ARDL method is applicable for long- and short-run estimation, even if the regressors are endogenous as the autocorrelation problem is eliminated and both the dependent and explanatory variables are well recognised and set apart. Therefore, ARDL modelling is adopted in this study, as follows:(5)$$\begin{array}{ll} \Delta LU5MR_{t}=\alpha_{0} & +\sum_{i=1}^{m}\alpha_{1\;}\Delta LU5MR_{t-i}+\sum_{i=0}^{n}\alpha_{2}\Delta LHE_{t-i}+\sum_{i=0}^{p}\alpha_{3}\Delta LGDPC_{t-i}\\ & +\sum_{i=0}^{q}\alpha_{4}\Delta LUEMP_{t-i}+\sum_{i=0}^{r}\alpha_{5}\Delta LUPOP_{t-i}+\lambda_{1}LU5MR_{t-1}\\ & +\lambda_{2}LHE_{t-1}+\lambda_{3}LGDPC_{t-1}+\lambda_{4}LUEMP_{t-1\;}+\lambda_{5}LUPOP_{t-1\;}\\ & +\mu_{t} \end{array}$$

Based on Equation (5), the long-run parameters are captured by $\lambda_{1}$ to $\lambda_{5}$, while the summations from $\alpha_{1\;}$ to $\alpha_{5}$ is related to short-run dynamics. The constant is denoted by $\alpha_{0}$, whereas Δ is the first difference operator and $\mu_{t}\;$is a white-noise disturbance term.

As a first step, the existence of long-run cointegration is examined using the ordinary least square (OLS) method and F-statistic value from Wald test, with the hypothesis, as follows:$$H_{0}:\;\lambda_{1}=\lambda_{2}=\lambda_{3}=\lambda_{4}=\lambda_{5}=0\;(\mathrm{No}\;\mathrm{cointegration})$$
$$H_{a}:\;\lambda_{1}\neq 0,\lambda_{2}\neq 0,\;\lambda_{3}\neq 0,\;\lambda_{4}\neq 0,\;\lambda_{5}\neq 0\;(\mathrm{There}\;\mathrm{is}\;\mathrm{cointegration})$$

Following Pesaran et al. [31] the null hypothesis, $H_{0}$, of no cointegration is rejected when the F-statistic value exceeds the upper bounds of critical value, thus the variables are cointegrated. However, if the F-statistic value is below the lower bound critical value, the variables are not cointegrated (fail to reject $H_{0}$). In summary, the F-statistic value is compared with the critical values of lower and upper bounds at different significant levels. The critical values suggested by Narayan [32] for sample size ranging between 30 and 80 cannot be used for this study, which is based on time series data of 22 years. Therefore, as an alternative specifically for small sample size analyses, the critical test values for this study were calculated using the response surface method, which Turner [33] extended from a prior analysis [34], in the following form:(6)$$C_{i}\left(p\right)=\beta_{0}+\frac{\beta_{1}}{T}+\frac{\beta_{2}}{T^{2}}+\varepsilon_{i}$$
where, $C_{i}\left(p\right)$ denotes the *p*% quantile estimate for the ith experiment, β values are based on response surface estimates expanded by Turner [33], T is the sample size and ε is a random walk.

Once cointegration is established, in the second step the coefficients of the long-run relationship of the model are generated through the Akaike information criterion (AIC) optimal lag structure. Subsequently, when there is evidence of a long-run relationship, as a third step, the short run ARDL model is estimated, based on re-parameterisation of long-run ARDL model as follows:(7)$$\begin{array}{ll} \Delta LU5MR_{t}=\alpha_{0\;} & +\sum_{i=1}^{m}\alpha_{1+i\;}\Delta LU5MR_{t-i}+\sum_{i=0}^{n}\alpha_{2+i}\Delta LHE_{t-i}\\ & +\sum_{i=0}^{p}\alpha_{3+i}\Delta LGDPC_{t-i}+\sum_{i=0}^{q}\alpha_{4+i}\Delta LUEMP_{t-i}\\ & +\sum_{i=0}^{r}\alpha_{5+i}\Delta LUPOP_{t-i}+ƴECT_{t-1} \end{array}$$
where, $ECT_{t-1}$ is the error correction term, which indicates the speed of adjustment back to long-run equilibrium after a short-run disturbance.

Finally, diagnostic and stability check of the models is undertaken to ascertain the goodness of fit of the ARDL model. The diagnostic test includes the Breuch–Pagan Lagrange multiplier (LM) test for autocorrelation, Jarque–Bera normality test, Breuch–Pagan–Godfrey (BPG) heteroscedasticity test. While the structural stability test is conducted by employing the cumulative residuals (CUSUM) and the cumulative sum of squares of recursive residuals (CUSUMSQ).

## 4. Results and Discussion

In employing the ARDL method, the stationarity test is initiated to test the order of integration for each variable, first using the Augmented Dickey–Fuller (ADF) and followed by Phillips–Perron (PP) unit root tests. Both ADF and PP unit root tests have a null hypothesis that the tested series has a unit root against the alternative of stationarity. This is to confirm that the regressors are I(0) and/or I(1) in order to proceed with the ARDL method. It is important to conduct the stationarity test to ensure that the variables are not integrated of order two, I(2), as this would lead to biassed and unreliable estimates.

Table 2 shows the results of the ADF and PP unit root test. Based on the result, the ADF shows that LU5MR and LUEMP are stationary at level (1% significant level) for both intercepts as well as the intercept and trend. Similarly, LUPOP at level is stationary for both the intercept (1% significant level) and the trend and intercept (10% significant level). While other variables, namely, LPUH, LPRH, LOOP, and LGDPC are insignificant at the level, thus, the ADF unit root test is conducted at the first difference. The result shows that LPRH, LOOP, and LGDPC are stationary (1% significant level) both at intercept as well as intercept and trend, while LPUH is stationarity (1% significant level) at intercept. Subsequently, the PP unit root test also reveals a similar mixture of stationarity for the variables at intercept as well as at intercept and trend, both at a level as well as at first difference. In conclusion, some of the model’s variables are I(1), while others are I(0), and none of the data series are I(2). This suggests that the ARDL bound testing procedure could be used to estimate the short and long-run relationships between variables.

The unit root test of this study has revealed that the dependent variable is stationary at level. According to Basu [35], because the dependent variable is stationary at level, spurious regressions may occur, and cointegration analysis may not be appropriate. Basu [35], on the other hand, employed the bounds testing procedure developed by Pesaran et al. [31], with F-statistics and critical values, to identify possibilities of long-run relationship. Alternatively, despite the fact that the requirements for ARDL and nonlinear ARDL are the same, Boulila [36] used a cointegration test for nonlinear ARDL analysis amidst the dependent variable being stationary at level. In fact, the requirement that the dependent variable should be I(1) is not largely acknowledged in the existing literature [37].

Therefore, the results of joint F-statistics, estimated based on Equation (5) through the OLS estimation procedure, and the calculated critical test values using the response surface method expanded by Turner [33], are shown in Table 3. There are three models, with different types of health expenditures, that have been estimated for cointegration. The first model is tested with PUH, the second model is PRH, and the third model is with OOP. Each model has another three explanatory variables, namely, GDPC, UEMP, and UPOP. The results reveal that F-statistics values are beyond the critical value at a 5% level of significance for Model 1, as well as at 1% significance level for Model 2 and Model 3. This confirms the existence of a long-run relationship among the variables.

Next, the long-run model is estimated based on Equation (5), following the ARDL cointegration technique for the long-run estimates. The empirical results of the long-run estimates are presented in Table 4. The results reveal that public health expenditure (Model 1) and private health expenditure (Model 2) are statistically insignificant. In Model 2, that is, when the model is tested with private health expenditure, GDP per-capita, and urban population, it is significant. In addition, only out-of-pocket health expenditure (Model 3) is statistically significant, which indicates that for every 1% increase in out-of-pocket health expenditure, it leads to an average of 0.61% increase in the under-five mortality rate. Similarly, in BRICS countries, Kulkarni [15] not only found a positive significance of out-of-pocket health expenditure on infant mortality rate, but also for public health expenditure, and implied that increases in health expenditure alone are not sufficient to improve health outcomes, unless they are accompanied by improvements in the financial and delivery system. While analysing the effectiveness of health expenditure in Sub-Saharan Africa, Ssozi and Amlani [22] revealed that overall total health expenditure improves child mortality, even though, separately, public and private health expenditure reduce under-five mortality rates and infant mortality rates, respectively. However, out-of-pocket health expenditure revealed a statistical insignificance in infant and child mortality rates. The findings of this study also do not concur with studies of similar nature, which focused on public and private health expenditures by Ashiabi et al. [23] and Boachie et al. [25]. Ashiabi et al. [23] revealed that public health expenditure was inversely and significantly related to infant and under-five mortalities, while private health expenditure was insignificant. While Boachie et al. [25] highlighted that public health expenditure lost its significance in improving child health outcomes when private health expenditure was included in the regression.

In fact, the findings of this study also contradict a previous study by Riayati and Junaidah [14] for Malaysia, which reported that there was no cointegration between the under-five mortality rate and public health expenditure. The distinction in findings could be due to differences in sample size and control variables. Probably, the 26 years of time series study by Riayati and Junaidah [14] could have yielded a similar outcome to this study, if F test critical values for cointegration were calculated using the response surface method expanded by Turner [33] for sample sizes below 30, rather than the critical values suggested by Narayan [32] for sample sizes ranging between 30 and 80. In addition, this study analysed macroeconomic variables, while Riayati and Junaidah [14] included governance-related variables.

As for the findings in this study for Malaysia, they reveal that higher out-of-pocket health expenditure is disastrous for child health. Largely, out-of-pocket health expenditure may be unexpected and impact other priority or necessary household expenses, which may subsequently impact health-seeking behaviour, probably by deferring medical care due to financial deprivation. This may cause health to deteriorate, and, in severe circumstances, it may cause death. The result does not suggest that out-of-pocket payments are the best way of financing healthcare to improve child health, and it implies that the affordability of out-of-pocket payments is a barrier to seeking medical care. As such, out-of-pocket payments for medical care need to be reduced as they are reversing and worsening the child’s health outcome. Therefore, a reliable and effective health financing system as well as a targeted health-related safety net are needed as protection to prevent catastrophic health expenditure. Policymakers may consider setting up a health welfare fund, a health-related voucher system, or pooled health insurance premiums, especially for those vulnerable communities coping with unbudgeted out-of-pocket health expenditure, as it is negatively affecting children under-five. This would also need to be supported by a regulatory framework to ensure the efficient allocation of funds. In addition, a sustainable balance among the sources of health financing needs to be achieved efficiently and technically to ascertain that the health financing mechanism is affordable and responsive to the health of the population, including children under the age of five.

Furthermore, only in Model 2, that is, when the model is tested with private health expenditure, GDP per-capita, and urban population, there is a significant impact. The GDP per-capita was found to be positively significant, which indicates that a 1% increase in GDP per-capita leads to about a 2.5% increase in the under-five mortality rate in Malaysia. Almost all the relevant literature reviewed in this study reported either a negative relationship [18,25] or no significance [8,9,11,13]. Except for Kulkarni [15], whereby a positive, significant relationship was revealed between GDP per-capita and infant mortality rate in an analysis among BRICS countries (Brazil, Russia, India, China, and South Africa). Probably, for Malaysia, when the economy is good, the lifestyle change may not have addressed health needs and may negatively impact the health status, especially for children under-five. However, the urban population is statistically significant in decreasing the child mortality rate, similar to a study in Nigeria [8] and in Sub-Saharan African countries [16]. A 1% increase in the urban population would decrease the under-five mortality rate in Malaysia by 6%. Largely, urbanisation is motivated by better opportunities, including the increased access to healthcare facilities, which may increase the chances of child survival.

Table 4 also presents short-run estimation. The result reveals that only private health expenditure, especially with lag 1, significantly decreases the under-five mortality rate. Furthermore, in the short run, GDP per-capita and the urban population were significant across all three models, while unemployment was only significant in Model 1.

Most importantly, the lagged error-correction term (ECT), which is represented by ECM(-1) in the result, is negative and statistically significant at the 1% level in all three models. In Model 1, even though the ECT coefficients explain that deviation from equilibrium in the current year will be corrected by 54% in the following year; however, the results reveal that in the long run, none of the variables are statistically significant. In Models 2 and 3, the ECT appears with a coefficient of −1.06 and −1.36, whereby, according to Narayan and Smyth [38], any ECT value between −1 and −2 implies that, rather than a direct convergence to equilibrium, the error correction process fluctuates around the long-run value in a dampening manner, and once the process is complete, convergence to long-run equilibrium will be relatively fast. Similar findings and justification have also been reported by Yoong et al. [39].

### Robustness Analysis

This analysis also includes the fully modified ordinary least squares (FMOLS) method and dynamic ordinary least squares (DOLS) method regression tests to account for the robustness of long-run parameters in order to ensure that the results achieved in the ARDL model are reliable [40,41,42]. The FMOLS method developed by Phillips and Hansen [43] is based on a semi-parametric approach to estimate the long-run parameter. It provides a consistent parameter in a relatively small sample, and also controls the possible endogeneity, serial correlation, and omitted variable bias, while allowing for heterogeneity in the long-run parameters. While the DOLS technique proposed by Stock and Watson [44] employs a parametric technique in estimating the long-run parameter in the model, DOLS also provides unbiased estimators while correcting the potential endogeneity issue.

The results of FMOLS and DOLS long-run estimation are as shown in Table 5. As for Model 1, both the FMOLS and DOLS methods revealed that $LPUH_{t}$ is statistically insignificant, similar to the findings of the ARDL method. Likewise, in Model 3, both the FMOLS and DOLS techniques indicated that $LOOP_{t}$ is statistically significant, which is similar to the ARDL method’s findings. However, for control variables, when both FMOLS and ARDL approaches found insignificant outcomes, the result of DOLS was significant, which implies that $LGDPC_{t}$ and $LUPOP_{t}$ both exhibited a negative and positive effect on $LU5MR_{t}$, respectively. As for Model 2, the DOLS method appears to support the ARDL technique’s insignificance of $LPRH_{t}$, although FMOLS obtained a significant result. Furthermore, the FMOLS validates the significance of $LUPOP_{t}$ as determined by the ARDL approach, whereas $LGDPC_{t}$ was insignificant in both the FMOLS and the DOLS. Basically, the findings related to control variables in Models 2 and 3 seem to be not robust as the statistical significance varies. Overall, although there are some differences in terms of the significance, the results of both the FMOLS and DOLS methods support the ARDL technique as the signs and magnitudes are similar. In fact, as Hundie [45] emphasized, the ARDL is more reliable in interpreting the long-run coefficients. Most importantly, all three methods provide the same evidence for $LOOP_{t}$, which confirms a significant worsening impact on the under-five mortality rate in Malaysia.

The results of relevant diagnostic tests are as shown in Table 6; the Breuch–Pagan Lagrange multiplier (LM) test for autocorrelation, the normality test by Jarque–Bera, and the heteroscedasticity test by Breuch–Pagan–Godfrey (BPG). These tests reveal that the models are free from autocorrelation and heteroskedasticity problems and indicate that the estimated residuals are normally distributed. In addition, the CUSUM and CUSUMSQ test results, as shown in Figure 4, were within the 5% level boundaries (represented by the two straight red lines), which suggests that the parameters are stable over the sample period. Even though the CUSUM test exceeded a little the two red lines in Model 1, but still, it is a reliable model.

## 5. Conclusions

This study investigates the relationship between public, private, and out-of-pocket health expenditures and the under-five mortality rate in Malaysia using the ARDL estimation technique. New critical test values are recalculated for a bound testing technique of cointegration using the response surface method expanded by Turner [33], as this is based on a small sample size of time series data of 22 years for the period 1997–2018. Furthermore, the FMOLS and DOLS methods are applied to check the robustness. Despite minor differences in significance, the results of both the FMOLS and DOLS methods corroborate the ARDL technique since the signs and magnitudes are almost similar, and most importantly, all three methods provide the same evidence for out-of-pocket health expenditure. The findings suggest that an increase in out-of-pocket health expenditure leads to an increase in the under-five mortality rate, which is a worsening state. While public and private health expenditures are statistically insignificant. Even though out-of-pocket health expenditure is unavoidable, public funding is scarce, and private health spending is market-driven, due consideration may be given to restructuring the health financing system. As an increasing out-of-pocket is disastrous to child health outcomes. As well, a policy reform could be mainly targeted at those vulnerable populations to ensure universal health coverage, including for children.

This study has its limitations. The published data on private and out-of-pocket health spending in Malaysia is only available for the past 22 years, but public health spending data is available for over 50 years. As a result, the analysis of this study was retained for 22 years for consistency purposes. As such, probable future research may perhaps focus on the total health expenditure, narrow the analysis within the out-of-pocket health expenditure, or perhaps expand the analysis on population health. As for other control variables, it has been revealed that GDP per-capita and urban population are significant only when the model is tested with private health expenditure. In conclusion, the study indicates that out-of-pocket health expenditure deteriorates the under-five mortality rate in Malaysia.

## Figures and Tables

**Figure 1 healthcare-10-00589-f001:**
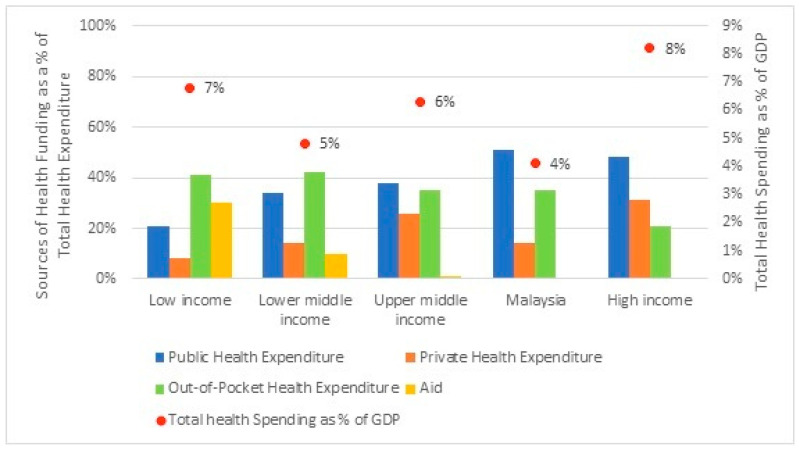
Global health expenditure in 2018. Source: Report on Global Spending on Health 2020: Weathering the Storm (WHO, 2020) and Malaysia National Health Accounts: Health Expenditure Report 1997–2018 (Ministry of Health, 2020).

**Figure 2 healthcare-10-00589-f002:**
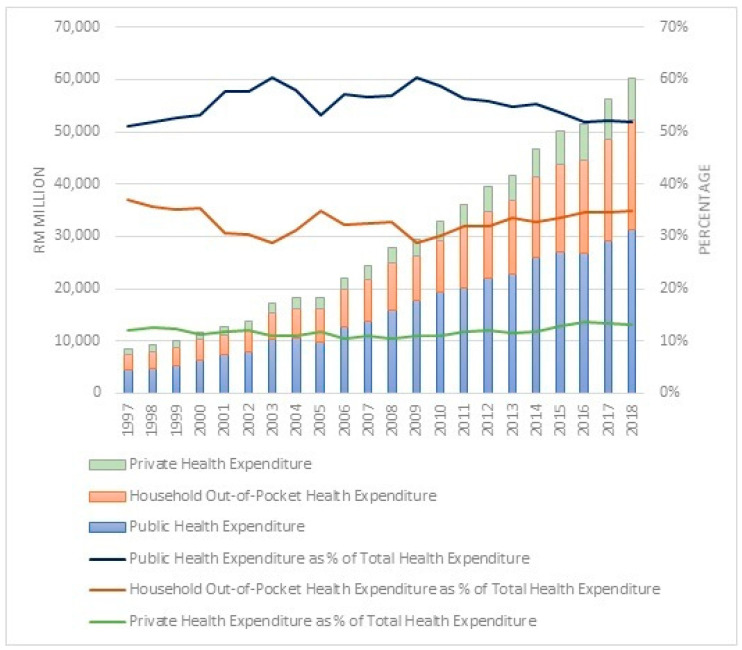
Health expenditure in Malaysia, from 1997 to 2018. Source: Malaysia National Health Accounts, Health Expenditure Report 1997–2018 (Ministry of Health, 2020).

**Figure 3 healthcare-10-00589-f003:**
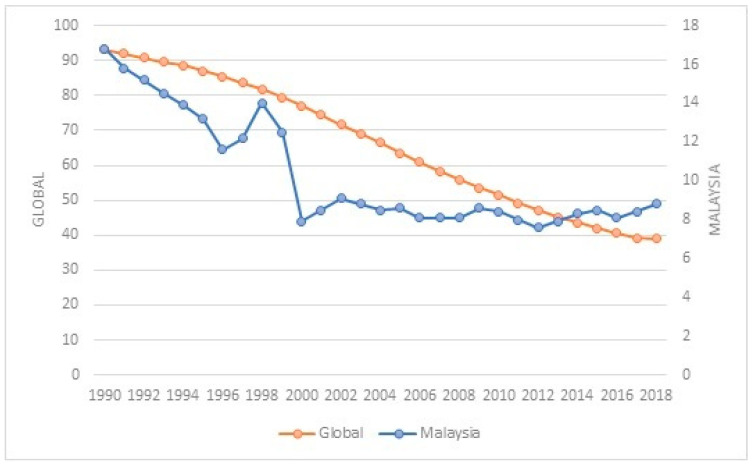
Under-five mortality rate from 1990 to 2018. Source: Department of Statistics Malaysia and United Nations Inter-Agency Group for Child Mortality Estimation 2018.

**Figure 4 healthcare-10-00589-f004:**
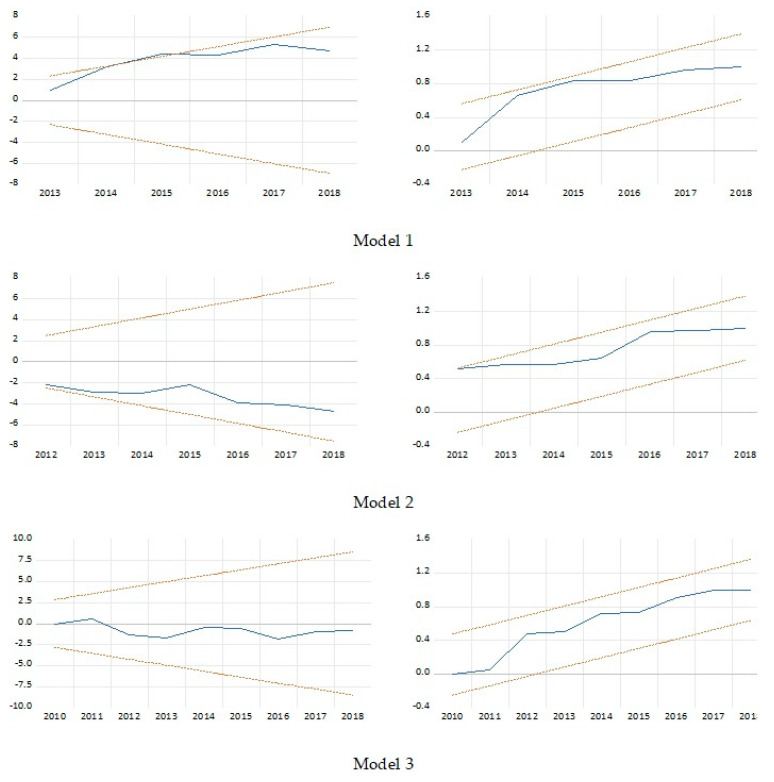
Cumulative sum (CUSUM) and CUSUM square test for stability.

**Table 1 healthcare-10-00589-t001:** Descriptive Statistics.

Variable	Mean	Median	Maximum	Minimum	Standard Deviation
Under-five mortality rate, U5MR	8.954545	8.45	14	7.6	1.668112
Public health expenditure, PUH	18,810.91	17,625	31,206	7882	7517.776
Private health expenditure, PRH	4090.994	3346.173	7925	1850.447	1964.397
Out-of-pocket health expenditure, OOP	11,210.65	10,082.24	21,016	5438.689	4929.542
Gross domestic product per-capita, GDPC	26.02288	26.14929	37.04429	16.43146	6.759924
Unemployment, UEMP	381.1091	369.15	504.3	214.9	72.99758
Urban population, UPOP	18,454,162	18,560,085	23,973,075	12,557,524	3,538,747

**Table 2 healthcare-10-00589-t002:** Augmented Dickey–Fuller and Phillips–Perron unit root test results.

	Augmented Dickey–Fuller		Phillips–Perron		
	Intercept	Intercept and Trend	Intercept	Intercept and Trend	Result
Level					
LU5MR	−9.2310 ***	−7.3005 ***	−3.1687 **	−1.6703	Stationary—I(0)
LPUH	−1.7555	−2.2980	−2.6512 *	−2.0188	Non-stationary
LPRH	0.3205	−2.2799	0.4604	−2.2670	Non-stationary
LOOP	0.7835	−2.6473	2.6126	−3.8490 **	Non-stationary
LGDPC	−0.2750	−3.0272	0.1375	−3.0755	Non-stationary
LUEMP	−2.9089 *	−5.3588 ***	−2.9253 *	−5.3215 ***	Stationary—I(0)
LUPOP	−4.4014 ***	−3.6129 *	−15.1749 ***	−2.7543	Stationary—I(0)
First Difference					
LU5MR					
LPUH	−4.3468 ***	−3.3031	−5.8003 ***	−14.1098 ***	Stationary—I(1)
LPRH	−5.2424 ***	−5.1365 ***	−5.2484 ***	−5.1851 ***	Stationary—I(1)
LOOP	−4.9086 ***	−4.9865 ***	−7.7188 ***	−8.2912 ***	Stationary—I(1)
LGDPC	−6.1636 ***	−5.9929 ***	−6.9335 ***	−7.4665 ***	Stationary—I(1)
LUEMP					
LUPOP					

Note: “L” denotes that data series are in logarithm form, while *** indicates the test statistic is significant at the 1% significance level, ** indicates 5% significance level, and * indicates 10% significance level.

**Table 3 healthcare-10-00589-t003:** Bound test (cointegration) results.

				ARDL	F-Statistic		Outcome
Dependent variable: LU5MR_t_					k = 4		
Independent variables:							
Model 1: LPUH_t_	LGDPC_t_	LUEMP_t_	LUPOP_t_	(2, 1, 2, 2, 2)	6.403994 **		Long-run relationship exist for Models 1, 2, and 3
Model 2: LPRH_t_	LGDPC_t_	LUEMP_t_	LUPOP_t_	(2, 1, 2, 0, 2)	14.54388 ***		
Model 3: LOOP_t_	LGDPC_t_	LUEMP_t_	LUPOP_t_	(2, 0, 2, 0, 2)	9.715326 ***		
				Critical Value	Lower Bound	Upper Bound	
				1%	6.3710	8.6932	
				5%	4.0581	5.6960	
				10%	3.1910	4.5631	

Note: Critical values are calculated using the response surface method expanded by Turner [33]. *** refers to the statistical significance level at 1%, and ** refers to the statistical significance level at the 5%.

**Table 4 healthcare-10-00589-t004:** Long-Run Estimates.

Dependent Variable: LU5MR_t_	Model 1		Model 2		Model 3	
Estimated Long-Run Coefficients						
LPUH_t_	1.1077					
LPRH_t_			0.3212			
LOOP_t_					0.6133	**
LGDPC_t_	8.6710		2.5040	**	1.0835	
LUEMP_t_	4.2447		1.0484		0.6010	
LUPOP_t_	−20.9637		−5.8768	**	−3.4945	
Constant	290.7363		83.8707		47.6196	
Estimated Short-Run Coefficients from Error Correction Model						
Δ LU5MR _t−1_	0.4588	**	0.6864	***	0.9010	***
Δ LPUH _t_	0.1168					
Δ LPRH _t_			0.2886			
Δ LPRH _t−1_			−0.8447	***		
Δ LGDPC _t_	0.7266		1.5041	***	0.2862	
Δ LGDPC _t−1_	−1.9109	***	−1.0773	***	−0.5665	**
Δ LUEMP _t_	0.9691	***				
Δ LUEMP _t−1_	−0.7119	**				
Δ LUPOP _t_	−65.3317	***	−58.6607	***	−45.6088	***
Δ LUPOP _t−1_	27.3004		41.8924	***	51.6903	***
Constant	156.0336		89.1833		64.7511	
ECM(-1)*	−0.5367	***	−1.0633	***	−1.3598	***

Note: *** and ** indicate the test statistic is significant at 1% and 5% significance level.

**Table 5 healthcare-10-00589-t005:** Long-run estimates based on FMOLS and DOLS method.

Dependent Variable:	Model 1		Model 2			Model 3			
LU5MR_t_	FMOLS	DOLS	FMOLS		DOLS	FMOLS		DOLS	
LPUH_t_	−0.3919	−0.5616							
LPRH_t_			0.7485	**	0.3940				
LOOP_t_						0.8799	**	0.8047	**
LGDPC_t_	0.8162	−4.8246	0.9368		−4.6485	−0.0283		−4.2504	**
LUEMP_t_	0.4592	−1.9908	0.0545		−2.5058	0.0555		−1.5910	
LUPOP_t_	−1.0453	11.7582	−3.6006	**	10.0059	−2.4655		8.4986	**
Constant	18.1246	−163.4274	52.8760		−140.5381	35.0300		−126.9745	

Note: ** indicate the test statistic is significant at 5% significance level.

**Table 6 healthcare-10-00589-t006:** Diagnostic test results.

	Model 1	Model 2	Model 3
LM test	1.1901	2.8584	2.1811
	(0.3930)	(0.1487)	(0.1835)
Normality	0.1244	1.0596	1.1758
	(0.9397)	(0.5887)	(0.5554)
BPG	1.1662	0.6380	0.8066
	(0.2754)	(0.7644)	(0.6216)

Note: The number in parentheses represents the *p* value for the respective coefficient

## Data Availability

The data presented in this study are openly available from the Malaysian Government agencies, namely, the Department of Statistics (https://www.dosm.gov.my, last accessed on 14 September 2021) and Ministry of Finance (https://www.mof.gov.my , last accessed on 16 September 2021), as well as World Bank Open Data repository (https://data.worldbank.org/indicator/SL.UEM.TOTL.ZS?locations=MY, last accessed on 15 September 2021).

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
