# Peer review of "The Impact of Public, Private, and Out-of-Pocket Health Expenditures on Under-Five Mortality in Malaysia"

_healthcare, 2022, doi:10.3390/healthcare10030589_

Round 1
Reviewer 1 Report
The authors described that the finding of this study contradicts with previous study. The contradiction might be caused by misspecification.
Please reconsider the cointegration relationship among macroeconomic variables you used.
・Unit roots tests
The authors summarized the results of unit roots tests at Table 2 where two data generating processes were assumed. One was a difference-stationary process, and another was a trend-stationary process.
[1] In general, researchers should do the stepwise test; for example, test the difference-stationarity after rejecting the trend-stationarity.
Please add your procedure of unit roots tests into the text.
In addition, please delete Outcome, and use Result in Table 2.
[2] When a level variable follows the difference-stationary process, the unit roots tests for the first difference series are meaningless.
Please delete some results of unit roots tests for the first difference series.
[3] Please clarify the sentence below.
unemployment are integrated of I(0) I(1), while other variables are integrated of I(1).
・Cointegration relationship
As shown in Table 2, LU5MR is considered as I(0). However, the authors wrote that the null hypothesis of no cointegration is rejected for all three Models.
[4] Because, in general, there is one cointegration relationship among I(1) variables, please cite the previous study regarding the specific cointegration relationship between I(0) variable (dependent variable) and I(1) variables (independent variables), and explain the characteristics of this cointegration relationship briefly.
・Specification
[5] Readers may want to know the reason why only Model 2 included LGDPCt, LUEMPt, and LUPOPt.
Why not Model 3 include these variables?
Please add these reasons into the text.
[6] Did the authors check a cointegration relationship in the right-hand side variables of Model 2?
If there is a cointegration relationship in the right-hand side variables of Model 2, the authors had better use the linear combination among these variables as an explanatory variable of LU5MR. The linear combination can be derived from the cointegration test. Please check.
[7] The sentence below shown in page 10 should be moved to previous page.
Furthermore, only in Model 2, that is when the model is tested with private health expenditure, GDP per capita and urban population is significant.
・Conclusions
In Conclusions, the authors argued that the results are stable and robust as all the models yield consistency result. Is it true?
[8] Result of Model 3 in Table 4 shows that LGDPCt had no effect on LU5MRt in the long run. In contrast, result of DOLS (Model 3) shown in Table 5 indicates that LGDPCt had a negative effect on LU5MRt at the 5% significance level.
Please describe the differences between two results.
Reviewer 2 Report
The Impact of Public, Private and Out-Of-Pocket Health Expenditures on Under-Five Mortality in Malaysia
Submitted to Healthcare (MDPI)
Review
Summary
The authors examine different types of healthcare expenditures in Malaysia and check their respective co-integrative influence on the mortality of under 5-year olds [U5MR] in that country. The latter dependent variable has been seen deteriorating over recent years. Using a response-surface method amended by Turner (2006) the authors find that all three healthcare-expenditure variables (public spending [PUH] / private spending (on insurance) [PRH] / out-of-pocket private spending [OOP]) do not reject the null-hypothesis of no co-integration. However, only the long-run impact of OOP on U5MR is statistically significant, with a 1% increase in OOP pointing to a 0.61% increase in U5MR. The authors conclude that “out-of-pocket health expenditure deteriorates under-five mortality rate[s] in Malaysia” (p. 14).
The paper reads well, and I find the statistical analysis credible. I believe that the paper can be published after a thorough English proofread and grammar check. The authors could also do a slightly better job at explaining how their main explanatory variables (PUH, PRH, OOP) fit in their model. They are introduced after Eq. (2) on p. 6, but the reader does not encounter them again until p. 9. In particular, I did not understand where Eq. (1) was ever used, what exactly are the components of X (“all the inputs that determine the health outcome” (p. 6) is too vague), and where the main explanatory variables fit in (2), (3), or (5). Here the authors could do a much better job in their exposition of the model.
Minor comments and typos
The paper contains many typos and it needs a thorough professional proofing. The following list is only indicative.
- Abstract: … is it “response-surface method” (without “s”)?! … Note that Turner (2006) only amends an earlier paper by Kanioura and Turner (2005). The general idea of response surfaces probably goes back at least to Box and Wilson (1951).
- P3L3: composition --> proportion
- P3L4: put footnote after the punctuation (full stop in this case). This repeats later in the paper, e.g., on P4L5.
- P3L8/9: death --> deaths (x2)
- P4L-3: sought --> seeks
- P5: in Nigeria … it was also found … --> delete “also”
- P5L-11: decrease --> decreases
- P6L2: Where else --> reformulate
- P6L12: per-capita income (with the dash); missing dashes of this kind appear frequently in this paper, e.g., with “long-run”. Please check carefully.
- P6L-1: refer --> refers
- P7L17: Therefore, the ARDL modelling --> Therefore, ARDL modelling
- P7L22: F-statistic (with the dash); this repeats (see also point 9 above)
- P7L-8: response-surface model by Turner (see also points 1 and 9 above)
- P8L2: as follow --> as follows
- P9L1: Equation (3) should be referred to in round parentheses (as it is not a literature reference. This type of typo repeats later … please check.
- P10L6-7: “… increase in the under-five mortality rate” :: The actual causality might of course run the other way, in the sense that a lack of insurance coverage prompts the parents to pay for emergency procedures trying the prevent the death of their young children. That is, a relatively inelastic demand for healthcare for the young prompts a high OOP. When viewed in relative terms (i.e., if PUH + PRH + OOP = 1), then an increase in PRH or PUH would tend to lower U5MR.
- P13L4: for a bound
- P13L1: Conclusions --> Conclusion // Overall, I find Section 5 at present the weakest section in the paper. It is rather political and thus feels a little as if the study was conducted to prove a point (which may of course have been true and that would be understandable). My recommendation would be to dial back the tone a little and stay close to the findings, and point to what else could be tested in the future. Then there is nothing wrong with the last sentence, except that I would replace “confirms” (which carries an undertone of something like “I told you so”) with “indicates” for example. Overall, I think there is a lot of room for improving the conclusion section; one can add spice, but it needs to be done with a lot of thought and moderation.
- P13L7: the FMOLS and DOLS methods
- P13L-3: “yield consistency result” :: this is unclear [also there should be a comma before “as” on that line)
- P14L3: add comma before “as”; to child --> to the child
- P14L-2: finding --> study
- In reference 4, why is there such a big space after “Monteiro”
- In reference 6, “well-being” (no extra space)
Reviewer 3 Report
- For the introduction, I feel it is too long, and some of the context information may be unnecessary. On page 4, line 8, “This indicates that Malaysia’s success will not only be measured in terms of growth and wealth, but also in the ability in ensuring the well-being of children in Malaysia”. This sentence looks abrupt. From what was stated before, I do not understand why success in Malaysia also relies on well-being of children.
- For methodology and Data, the data subsection (3.1) could be moved before the method section as it is just one paragraph. How large is the sample size? The author just mentioned a small sample size, but it is better to give the exact number here.
In addition, when the author introduced the covariates such as “Gross Domestic Product per capita (????), unemployment (????), and urban population (????)”, are these variables normally controlled in this type of studies, or they were chosen based from the previous empirical studies? If so, the citation should be put here for reasoning the selection of those covariates.
On page 6, paragraph 5, the author wrote “the ?? will be represented by the public (???), private (???), and out-of-pocket (???) health expenditures separately.” Could the author further explain why HE should be represented in this way?
- For the results section, it is better to report with some numbers for significant tests in the text.
- For the conclusion section, what is the limitation of the current study? It is better to add some limitations to the current study.
Round 2
Reviewer 1 Report
No systematic discussion regarding point 6 can be found.
If there is no misspecification, we believe that the authors can give explanation about the characteristics of the cointegration relationship between I(0) variable and I(1) variables.
I strongly recommend the authors to cite the previous study.
"The assessment has passed the bound test" is not enough.
Reviewer 3 Report
This revision looks much better! I think it is ready to go for a publication
Author Response
Thank you. . We appreciate the time and effort that you have dedicated in providing valuable feedback on our manuscript.